# Reassessing the Environmental Kuznets Curve in Relation to Energy Efficiency and Economic Growth

**Jie Zhang [1], Majed Alharthi [2], Qaiser Abbas [3,*], Weiqing Li [4,*], Muhammad Mohsin [5], Khan Jamal [6] and Farhad Taghizadeh-Hesary [7]**

[1] School of Economics and Resource Management, Beijing Normal University, Beijing 100875, China; zhangjie0190@hotmail.com

[2] Finance Department, College of Business, King Abdulaziz University, P.O. BOX 344, Rabigh 21911, Saudi Arabia; mdalharthi@kau.edu.sa

[3] Department of Economics, Ghazi University, Dera Ghazi Khan 32200, Pakistan

[4] School of International Economics and Tourism Management, Zhejiang International Studies University, Hangzhou 310023, China

[5] School of Finance and Economics, Jiangsu University, Zhenjiang 212013, China; m.mohsin3801@yahoo.com

[6] Institute for Region and Urban-Rural Development, Wuhan University, Wuhan 430072, China; jamalkhan_87@yahoo.com

[7] Social Science Research Institute, Tokai University, Hiratsuka 259–1292, Kanagawa, Japan; farhad@tsc.u-tokai.ac.jp

* Correspondence: Qabbas@gudgk.edu.pk (Q.A.); 88924381@163.com (W.L.)

**Abstract:** Energy consumption and its efficiency are significant factors for economic growth and environmental stress. This study postulates the occurrence of the Environmental Kuznets Curve hypothesis (EKC) by using the Autoregressive-Distributed Lag (ARDL) model. Furthermore, a data envelopment analysis (DEA) model is used to measure energy efficiency, energy intensity, and environment to view the trajectory of EKC for the underline economies. For this purpose, a panel dataset from 1990–2013 of 15 developing countries is analyzed to verify the objectives mentioned above. The results of the panel ARDL support EKC's theory for underline economies, as GDP positively impacts carbon emissions, while the square of GDP is negatively related. The DEA-based results found relatively low environmental conditions in these emerging economies due to high energy intensity and low energy efficiency. This outcome suggests that renewable energy sources must be treated as an essential factor for achieving sustainable economic goals without environmental degradation.

**Keywords:** environmental Kuznets curve; $CO_2$ emission; energy efficiency; economic growth; panel ARDL; DEA

## 1. Introduction

Climate change is a devastating phenomenon that people have experienced for the last few decades. Excessive greenhouse gases (GHGs) in the atmosphere, specifically nitrous oxide ($N_2O$), carbon dioxide ($CO_2$), methane ($CH_4$), and chlorofluorocarbons (CFC), are the major causes of global climate change [1,2]. This phenomenon will cause a dramatic change in our world in the coming years, as greenhouse gases absorb heat from the sun and capture it in the atmosphere, causing the Earth's surface temperature to rise. The fifth assessment report of the International Panel on Climate Change (IPCC) concluded that the climate system's human impact is visible [3]. After the Industrial Revolution, population growth and economic development have led to an increase in greenhouse gas emissions and climate because of human interference on the Earth. Among the several greenhouse gases that cause global climate change, carbon dioxide is the most abundant [4]. Its excess may be directly associated

with human activities, such as burning fossil fuels, transportation, deforestation, land reclamation, and cement production for agricultural purposes, which increased after the industrial revolution.

In this modern age, enhanced development is the key to the progress of each nation. Developing nations focus on their growth process to boost productivity and grow early. These nations are also attempting to improve their living standards by raising their per capita incomes, and this is being made possible with the help of enhanced growth and development. Emerging economies such as developing Asian nations are now relying on industrialization for their rapid development. In this regard, these developing Asian economies rely on the rapid productivity energy cycle to attain the desired economic upswing. The environment is being affected by $CO_2$ emissions due to the energy process, as industrialization had led to environmental degradation. Thus, the swift economic upswing gives rise to environmental depletion in developing economies [5]. These Asian economies have been trying to achieve the desired economic upswing for the last two decades but have failed to clean their natural environment. Inadequate policies and limited resources fail to coordinate with the harmful environment, which is causing ecological disorder. Therefore, the developing Asian nations have compromised their environmental conditions and focus on their per capita incomes to increase the so-called standard of living. Therefore, it can be rightly claimed that ecological disorder is rising with the growing rate of economic upswing, which is reflected in the Environmental Kuznets Curve (EKC) [6].

The world energy consumption crossed BTU 583.57 quadrillions in 2017, at a 2.27 percent annual growth rate, against BTU 381.49 quadrillions in 1998. The share of underline developing economies crossed BTU 213.96 quadrillions in 2017, with thirty-seven percent of the world's energy consumption [7]. Thus, energy consumption has increased considerably in these developing economies, with swift economic growth as the desired output. However, environmental degradation has a peak off as the undesired output due to dirty energy sources such as fossil fuel. Most of the energy sources of the underline countries are import-based, which is hurting their economic progress and exchange rate. The world is turning the sources of energy into renewable ones, with 570.96 million tons of renewable energy consumed. However, the share of developing economies is less than twenty-five percent [8].

The current COVID-19 condition has changed the economic and energy scenario. Renewable energy projects have been delayed in developing economies to meet the current financial requirements. The oil price fluctuations during COVID-19 and its impact on the exchange rate have opened a new debate on energy efficiency and economic growth. In this case, the occurrence of EKC theory and its smooth trajectory is debatable, as economic growth is not the only independent factor responsible for it.

There were nearly 20.06 billion units of gross fixed capital formation worldwide in 2018. The underline economies hold 7.77 billion, which is nearly thirty-nine percent of the total [9]. Unfortunately, these developing economies accumulated capital, which is not a technological advance and is thus less efficient in production. It is not only a source of high per-unit cost but can also be a source of environmental degradation due to the high usage of fossil fuel energy, which is imported and dirty.

Developing Asian nations have specific problems, like poverty, unemployment, and a high population growth rate with a low per capita income growth rate. That is why these nations try to boost their development to resolve these issues. However, due to the scarcity of resources and outdated technology, it is difficult to control environmental degradation. In per capita terms, China is responsible for 7.95 metric tons of emissions in 2018 compared to 2.69 metric tons of emissions in 1999, at an annual growth rate of 5.97 percent [10]. Likewise, Malaysia faced 8.02 metric tons of emissions in 2013 compared to 7.76 metric tons in 2010. Finally, Mongolia recorded 14.54 metric tons of emissions in 2013 compared to 9.09 metric tons in 2010. Mongolia is considered the largest carbon dioxide emitter among developing countries in the Asian region [11]. Energy is regarded as the primary source of development, but it is essential not to consider improper planning, scarce resources, and outdated technology. Therefore, to control $CO_2$ emissions, it is essential to apply the policies about utilizing

energy sources. Governments and policymakers need better strategies to use energy efficiently to boost economic activity and control carbon dioxide emissions, especially in developing economies [12].

The current work includes the data of those developing Asian nations that share common social issues. They are also on the same page regarding geographical, financial, political, and ecological circumstances with a higher population growth rate. They are considered highly ranked as carbon dioxide emitters since the 1990s due to their will to become industrialized as quickly as possible. The panel Autoregressive-Distributed Lag (ARDL) and Data Envelopment Analysis (DEA) techniques were utilized to analyze the under-considered data for the following objectives:

1.  Reassessing the occurrence of EKC by observing the influence of economic upswing on carbon emission evolving Asian economies.
2.  Calculating the association between energy (renewable-renewable) and carbon emission.
3.  Examining the role of capital formation in $CO_2$ emissions and finally providing a policy to overcome the environmental challenges emerging due to high carbon emission.
4.  Measuring energy efficiency, energy intensity, and environmental conditions of the underlined economies by using DEA.

The analysis is organized as follows: the remainder of the introduction depicts the literature review, section two describes the materials and methods, results are provided in section three, discussions are in section four, and the conclusion is given in section five.

*The Review of Literature*

Research is conducted to evaluate the correlation between environmental issues and macroeconomic variables in recent decades. Numerous studies regarding this issue reviewed to verify the influence of energy usage and economic upswing on the developing economies' ecological disorder. Most of the studies focused on a panel of mixed-income nations such as upper income, middle-income, and lower-middle-income; few studies focused on regions such as the Association of Southeast Asian Nations (ASEAN) [13], Organization for Economic Co-operation and Development (OECD), and Organization of the Petroleum Exporting Countries (OPEC) [14]. In contrast, the panel of developing nations from the overall world was also a part of this literature [15], while Ref. [16] verified EKC was part of the developing one belt one road initiative. This literature review showed that energy usage and economic upswing positively correlate with the ecological disorder [17].

The Environmental Kuznets Curve hypothesis asserts a definite link among population growth, GDP, energy usage, and carbon emission. The positive association among these variables verified a specific rise in carbon dioxide emission when economies are developing. As the developing economies are in the development phase, EKC's presence witnessed and confirmed the constructive outcome of an economic upswing on the ecological disorder [18]. The authors [19] also proved the existence of the EKC hypothesis in thirteen nations by analyzing the association of eighteen economic indicators and ecological disorder. Moreover, population growth and economic uncertainty provide an essential answer to all the aforementioned variables for environmental depletion [20].

The authors of [21] verified the EKC theory's occurrence and demonstrated a negative effect of renewable energy on the environment. For the possible occurrence of EKC, environmental efficiency is very vital. It could be attained through energy efficiency, energy pricing, energy intensity, technological innovation, or building high-tech industries. The authors of [22] investigated the role of efficiency growth and convergence to enhance economic productivity using inputs and embrace technologies in 104 countries for a thirty-six-year dataset. This study found that environmental efficiency improved approximately 1.3 percent globally due to energy pricing, restructuring industrial setup, or globalization. The authors of [23] suggested that energy transactions can help to enhance economic and environmental efficiency. The transaction of energy in the different income level nations is different, which may change EKC's pattern or speed. The authors of [24] found that even global crises such as COVID-19 have changed the entire energy pricing mechanism and lead to the collapse of the energy market and the

competitiveness of renewable energy projects. Likewise, to view the EKC occurrence, the technological innovations in emission reduction and carbon transfer strategies based on the low carbon preference are deemed necessary. Ref. [25] suggested that low carbon preference can be an excellent source to improve environmental conditions without compromising economic growth. Some of the researchers attempted to verify the notion of EKC through capital formation. For example, Ref. [26] found that capital formation is a source of environmental degradation in G-7 countries. Therefore, the role of energy consumption, fossil fuel or renewable energy preferences, energy innovations to enhance the energy efficiency, or reduce its intensity, innovations for high-tech industrialization are a few core indicators to decide EKC's time and speed.

The existing literature is divided into two different aspects. Many attempted to view the EKC occurrence in developing or developed. For example, Ref. [27–29] analyzed the EKC theory without considering EKC's trajectory in routine or emergency conditions. Other studies like [30–32] attempted to measure the economic, energy, and environmental efficiency via indexing the variables of said field. These studies did nothing for EKC theory and its speed of occurrence.

In conclusion, some studies presented the assenting linkages of an economic upswing with carbon emission, while others delivered a negative association between these variables. The same is the case for energy, carbon emission, and economic upswing. Much of the research work evidenced the EKC hypothesis and established a panel of developed and developing economies. However, only a few have tried to fix the three-dimensional energy effect on economic growth and environmental stress. Moreover, the EKC literature focused on GDP and the conversion of GDP square term but not considering the other variables, such as capital formation, growth rate, and renewable energy consumption. Although these two also have an independent effect on economic growth and environmental condition, they can play an essential role in EKC trajectory and speed. Some other studies attempted to measure energy efficiency in economic cost and environment [33,34]. However, they did not explain EKC to view the real impact of energy efficiency on economic growth and the environment. Thus, there is a gap for some comprehensive studies in these areas, especially from developing nations of Asia. These nations are suffering much in terms of ecological disorder, energy usage, and sustainable economic growth. This research attempts to cover these two different concepts. This study attempts to fill the literature gap regarding energy efficiency as the source of the EKC trajectory. Here, the combined effect of energy, economic, and environment underlined developing economies' analysis to understand the EKC trajectory and speed. The study's efficiency score indicates the current condition of energy efficiency, energy intensity, and environmental efficiency of the individual country based on the last twenty-three years of progress to depict the gap of EKC among underline nations. Thus, lower-middle-income and upper-middle-income countries have been selected to view their respective economic and environmental conditions with energy efficiency as per world bank classification. Therefore, the current study has novelty because of its sole combination of variables, two different angels of analysis, and the selection of nations from the Asian region concerning their income levels. This study can help policymakers, and business individuals decide the course of EKC occurrence and its trajectory for preferring the supportive sources of renewable energy with high efficiency and low intensity in their respective countries.

## 2. Materials and Methods

A twenty-three-year panel dataset of fifteen developing economies of Asia was taken from 1990 to 2013. The primary source of this dataset is "World Development Indicators." This dataset has been divided into two income categories classified by World bank 2021. Here, Nepal, Bangladesh, India, Mongolia, Pakistan, Philippine, Vietnam, and Sri Lanka sorted out as lower-middle-income economies. At the same time, China, Iran, Jordan, Malaysia, Thailand, Turkey, and Indonesia are upper-middle-income economies [33]. This study is based on two different methods of research. First, the ARDL method of econometrics utilizes the EKC theory of environment and economic growth. Secondly, the DEA method of operational research to assess the energy efficiency of underline countries.

Table 1 depicts the detail of the indicators used for this study. Here, ecological disorder (END) is the dependent variable, while renewable energy (ENC), economic growth (EGW), the square of economic growth ($EGW^2$), capital formation (FCF), and population growth (PG) are the independent variables.

**Table 1.** Variable description and source.

| Description of Variables | Abbreviation | Unite | Source |
|---|---|---|---|
| Ecological disorder | END | Metric ton | WDI |
| Renewable energy | ENC | kg of oil equivalent | WDI |
| Economic Growth | EGW | GDP per capita | WDI |
| Square of Economic Growth | $EGW^2$ | GDP-square per capita (real term) | WDI |
| Capital Formation | FCF | Annual growth rate | WDI |
| Population Growth | PG | Growth rate | WDI |

### 2.1. Methodological Framework of ARDL

The model's technical specification is that the economic upswing and ecological disorder are positively associated in initial stages, whereas the square of GDP helps to reduce environmental depletion. The linear-quadratic equation confirms the presence of an inverted U-shaped EKC [34] and [35]. It can be written as:

$$END_{it} = \beta_0 + \beta_1 EN_{it} + \beta_2 EGW_{it} + \beta_3 EGW^2_{it} + \beta_4 FCF_{it} + \beta_5 PG_{it} + \mu_i \tag{1}$$

Equation (1) illustrates the linear quadratic equation to run the ARDL for the confirmation of EKC. Following the footprints of [36], Equation (2) establishes to assess the short-run ARDL results.

$$
\begin{aligned}
\Delta END_{it} = \beta_0 \quad + \quad & \sum_{i=1}^{k} \gamma_1 \Delta END_{i\,t-1} + \sum_{i=0}^{k} \alpha_1 \Delta END_{i\,t-1} + \sum_{i=0}^{k} \alpha_2 \Delta GW_{i\,t-1} \\
+ & \sum_{i=0}^{k} \alpha_3 \Delta GW^2_{i\,t-1} + \sum_{i=0}^{k} \alpha_4 \Delta FCF_{i\,t-1} + \sum_{i=0}^{k} \alpha_5 \Delta PG_{i\,t-1} \\
+ & \beta_1 ENC_{i\,t-1} + \beta_2 GW_{i\,t-1} + \beta_3 GW^2_{i\,t-1} + \beta_4 FCF_{i\,t-1} \\
+ & \beta_5 PG_{i\,t-1} + \mu_{it}
\end{aligned} \tag{2}
$$

In the above equation, $\Delta$ represents the difference, whereas $t-1$ used for cross-section shows the model's previous years. The $\alpha$ and $\beta$ are the coefficients of underline indicators. In the next step, the Error Correction Model (ECM) develops by formulating the following equation.

$$
\begin{aligned}
\Delta END_{it} = \quad & \beta_0 + \sum_{i=1}^{k} \gamma_1 \Delta END_{i\,t-1} + \sum_{i=1}^{k} \alpha_1 \Delta ENC_{i\,t-1} + \sum_{i=1}^{k} \alpha_2 \Delta GW_{i\,t-1} + \\
& \sum_{i=1}^{k} \alpha_3 \Delta GW^2_{i\,t-1} + \sum_{i=1}^{k} \alpha_4 \Delta FCF_{i\,t-1} + \sum_{i=1}^{k} \alpha_5 \Delta PG_{i\,t-1} + \\
& \beta_1 ENC_{i\,t} + \beta_2 GW_{i\,t} + \beta_3 GW^2_{i\,t} + \beta_4 FCF_{i\,t} + \beta_5 PG_{i\,t} + \delta ECM_{i\,t} + \\
& u_{it}
\end{aligned} \tag{3}
$$

The coefficient of ECM' $\delta'$ demonstrates the speed of adjustment, and it should be with a negative sign to show the convergence towards the long run from the short run to achieve the equilibrium condition.

### 2.2. Hybrid Error Correction Model

The Error Correction Model might show an error correction of the first difference exclusively, which is as follows:

$$\Delta Y_t = Y_t - Y_{t-1} \tag{4}$$

Error Correction Model can use for quantitative computation, and it is necessary to point out that it is the base of the Auto Regressive Distributive Lag Model (ARDL). We have used the Error Correction Model to condition that, if the ARDL sum coefficient is equal to 1, by decreasing the constant terms. Consequently, the coefficient of error correction term long-run association can attain if and only if the transformation at term grows at the constant rate, N. Hence, the coefficient mathematical model Error Correction Model can be presented as:

$$Y_t = \beta_0 + \beta_1 Y_{t-1} + \beta_2 Z_t + \beta_3 Z_{t-1} + \vartheta_t \tag{5}$$

The proposed term $Y_{t-1}$ is deducted from the ARDL both sides:

$$Y_t - Y_{t-1} = \beta_0 + \beta_1 Y_{t-1} + \beta_2 Z_t + \beta_3 Z_{t-1} - Y_{t-1} + \vartheta_t \tag{6}$$

$$\Delta Y_t = \beta_0 + \beta_1 Y_{t-1} + \beta_2 Z_t + \beta_3 Z_{t-1} - Y_{t-1} + \vartheta_t \tag{7}$$

Through addition and deducting $\beta_2 Z_{t-1}$ in the right-hand side of the mathematical model. The new equation is as follows:

$$\Delta Y_t = \beta_0 + \beta_1 Y_{t-1} + \beta_2 Z_t - \beta_2 Z_{t-1} + \beta_3 Z_{t-1} - Y_{t-1} + \beta_2 Z_{t-1} + \vartheta_t \tag{8}$$

or

$$\Delta Y_t = \beta_0 + (\beta_1 - 1)Y_{t-1} + \beta_2 Z_t + (\beta_2 + \beta_3)Z_{t-1} + \vartheta_t \tag{9}$$

To fulfill the condition of the Error Correction Model, the coefficient $(Z_{t-1})$ must be analogous to the deducted coefficient $Y_{t-1}$. So, the newly construed mathematical model is as follows:

$$\beta_1 - 1 = -(\beta_2 - \beta_3) \tag{10}$$

$$\beta_1 + \beta_2 + \beta_3 = 1 \tag{11}$$

Consequently, the term of error correction constant considers as:

$$\Delta Y_t = \beta_0 + \beta_2 Z_t - \tau(Y_{t-1} - Z_{t-1}) + \mu_t \tag{12}$$

$$\tau = -(\beta_1 - 1) = (\beta_2 + \beta_3) \tag{13}$$

If the variation in constant term increases at a continuous rate N, the association of the long-run phenomena is:

$$N = \beta_0 + \beta_2 N - \tau(y^* - Z^*) \tag{14}$$

$$\tau(y^* - Z^*) = \beta_0 + (\beta_2 - 1)N \tag{15}$$

$$y^* = \beta_0 + \frac{(\beta_2 - 1)N}{\tau} + Z^* \tag{16}$$

then the original order having and without having the value of log is considered as:

$$y_t^* = KZ_t^* \tag{17}$$

If we take the log from both sides, then it will be as:

$$Log y_t^* = Log K + log Z_t^* \tag{18}$$

Through using the anti-log of the new model, the long run will consider:

$$y^* = \exp\left[\frac{\beta_0 + (\beta_2 - 1)B}{\tau}\right] \tag{19}$$

where K represents the association between the variable Y and Z, the Error Correction Model is being used to measure the long-run relationship, and the Error Correction Model characterizes the previous imbalance in an existing factor. It can be:

$$\Delta N_t = \sum_{i=1}^{N} \tau_1 \Delta N_{t-1} + \tau_2 \Delta N_{t-1} \beta_2 N + YZ_t + \mu_t \tag{20}$$

## 2.3. Model Specification of DEA

DEA (Data Envelopment Analysis) is one of the many techniques for efficiency assessment. However, there are the following advantages to use the DEA method.

- Simultaneous analysis of outputs and inputs
- It is not necessary, a Priori, to define the frontier form
- Relative efficiency compared to the best observation
- Need no information on price

Let $V_k = (v_{ki}, \ldots, v_{kn})$ be the set of "n" environmental variables aimed at entity $k = 1, \ldots, K$. Environmental index (EVI) develops through underlying variables for each entity. Ranking the environmental performance of various entities is a general practice to develop an environmental index. It can differentiate with a choice of ordering $(\geqslant)$ defined on Rn. Therefore, the EVI can demonstrate through a mapping function such that $I : R^n \rightarrow R$, which satisfies

$$V_k \geqslant V_l \Leftrightarrow I(V_k) \geq I(V_l) \forall k, l \in \{1, \ldots, K\} \tag{21}$$

The assessment of each fundamental variable, which can represent through the function of the transformation unit, may be improved $F = (f_1, \ldots, f_n)$ such that

$$F : (V_{k1}, \ldots, V_{kn}) \rightarrow (f_1(V_{k1}), \ldots, f_n(V_{kn})) \tag{22}$$

As pointed out by [37] and [38], an admissible transformation engages extension and translation in such a way that $f_i(vk_i) = \alpha_i v_{ki} + \beta_1$, $\alpha_i > 0$. In correspondence with EVI, the order of various underlying entities which are expected to be chosen as inconsistent and associated with any acceptable conversion and transformation of fundamental factors being assessed in construction of EVI.

$$V_k \geqslant V_l \Leftrightarrow F(V_k) \geq F(V_l) \forall k, l \in \{1, \ldots, K\} \tag{23}$$

The geometric mean proved to choose with a useful index with strictly positive and ratio-scale variables—criteria for information loss in the direction of alternate combination techniques designed to develop indices [39]. The author of [40] used a non-compensatory approach of aggregation and discussed its usefulness. A nonparametric DEA methodology identifies a good frontier practice using a linear programming approach. It measures the comparative efficiency of underlying indicators based on outputs and inputs from comparable and measurable entities [41]. The DEA study by [42,43] used to measure the energy system performance, environmental performance, and productivity of different entities or decision-making units.

DEA's traditional use to measure environmental performance takes the difference between a good and a lousy output. For performance assessment, [42] introduced a fundamental academic foundation, which was the reason for the nonparametric DEA frontier practice's popularity to measure the wrong outputs. The vector $V_k = (v_{ki}, \ldots, v_{kn})$ is replaced by $X_k, \ldots, Y_k = X_{k1}, \ldots, X_{km}, Y_{k1}, \ldots, Y_{ks}$ (xk1,⋯, xkm, yk1,⋯, yks) to differentiate between inputs and outputs, where $X_k$ and $Y_k$ are input and output vectors, respectively. The input vector $X_k = (X_{k1}, \ldots, X_{km})$ is used to produce the output vector

$Y_{k\cdot} = (Y_{k1}, \ldots, Y_{ks})$. The inputs are $X \in R_+^p$ and the outputs are $Y \in R_+$. As a result, the production is the set of a potential combination of inputs and outputs:

$$
\begin{aligned}
S = \Big\{ (X,Y) : S &= \sum_{k=1}^{K} x_{ik} z_k \leq x_i, \; i = 1, \ldots, m \\
S &= \sum_{k=1}^{K} y_{rk} z_k \leq x_r, \; r = 1, \ldots, S \\
S &= \sum_{k=1}^{K} z_k = 1 \; i = 1, \ldots, m \\
z_k & 0, \; k = 1, \ldots, K \Big\}
\end{aligned}
\tag{24}
$$

In Equation (24) with constraint, the range-adjusted DEA model can be as follows:

$$
\begin{aligned}
\max \; &\frac{1}{m+s}\left( \sum_{k=1}^{K} \frac{S_i^-}{R_i^-} + \sum_{k=1}^{K} \frac{S_r^+}{R_r^+} \right) \\
S = \sum_{k=1}^{K} x_{ik} z_k &+ S_i^- = x_{0i}, \; i = 1, \ldots, m \\
S = \sum_{k=1}^{K} y_{rk} z_k &- S_r^- = y_{0r}, \; r = 1, \ldots, S \\
S = \sum_{k=1}^{K} z_k &= 1 \; i = 1, \ldots, m \\
z_k 0, \; S_i^- \; 0, &\; S_r^- \; 0.
\end{aligned}
\tag{25}
$$

the $x_{oi}$ is the i-th input and $y_{or}$ is the r-th output for entity $o(o) \in \{1, \ldots, K\}$; $R_i^-$ and $R_r^+$ show the ranges for output $r$ and input $i$, which can be defined as:

$$
R_i^- = max\{x_{ki}, k = 1, \ldots, K\} - min\{x_{ki}, k = 1, \ldots, K\} d
$$

$R_i^+ = max\{y_{ki}, k = 1, \ldots, K\} - min\{y_{ki}, k = 1, \ldots, K\}$ The additive DEA model's objective function is the inefficiency measurement of the entity's slack-based values, which can use to measure energy efficiency. The constraints decide the maximum possible reduction form the maximum reduction and the recognized extension in inputs and outputs. A variable having zero to one shows that all the entities have zero value to exclude in EVI. It is necessary to separate the relevant constituent in the objective function (Equation (25) while an equivalent restraint is required. The final constraint $\sum_{k=1}^{K} z_k 1$ appears to be a convexity situation, ensuring that the ratio-scale measurement units do not vary in the objective function. Any permissible conversion for the original factors, $f_i(vk_i) = \alpha_i k_i + \beta_1$ major attention on slacks accommodate the shifting parameter and the scaling factor $\alpha_i$ can be controlled through the adjustment range. After getting the optimal solution form Equation (25), the environmental index can define as:

$$
EI(v_0) = EI(X_0, Y_0) = 1 - \frac{1}{m+s}\left( \sum_{k=1}^{K} \frac{S_i^{*-}}{R_i^-} + \sum_{k=1}^{K} \frac{S_r^{*+}}{R_r^+} \right)
\tag{26}
$$

* shows the variable of consistent optimum slack. The EI derived from Equation (26) satisfies the following properties [44]:

P1. $0 \leq EI \geq 1$;
P2. EI (V0) = 1 ⇔ Entity o is situated on the best practice frontier;
P3. EI (V0) is inconsistent with the measurement units of outputs and inputs;
P4. EI (V0) contains the properties of strongly monotonic;
P5. EI (V0) is a conversion invariant.

P1 represents that Equation (26) provides a standardized index between 0 and 1, while the higher values are associated with better performance.

P2 shows that the underlying entities are essential to developing the best frontier practice with index values less than 1. It can see from Equation (27) that the identification of underlying entities determines the frontier of best practice associated with non-zero Zk, which are determined by the employed model.

P3 shows that the EVI index values are invariant through the ratio of scale dimension variables.

P4 demonstrates that a decrease in any input or any output causes the highest index value.

P5 translates that the addition and subtraction of constants through any variables do not impact indexes' values, especially after the interval-scale factors encompassed in constructing EVI [45]. In Equation (27), the normalization of linear min-max is accepted, while the normalized weighted version of underlying factors for all the entities is not. Indeed, the mentioned practice makes Equation (27) feasible, resulting in the ease of assessing the environmental index. However, [46] highlighted that the weighted sum aggregation rule presupposed the full compensability among underlying variables that are fully replaceable with each other [47]. Since various dimensions of underlying variables are not entirely replaceable with each other, the assumption may not be appropriate for measuring an environmental efficiency index [48].

$$
\begin{aligned}
\text{EVI}(V_k) &= 1 - \frac{1}{m+s}\left[\sum_{i=1}^{m}\frac{x_{ki}-min_k\{x_{ki}\}}{R_i^-} + \sum_{r=1}^{S}\frac{min_k\{y_{kr}\}-y_{kr}}{R_{r+}}\right] \\
\text{EVI}(V_k) &= \sum_{i=1}^{m}\frac{1}{m+S}\left[\frac{max\{x_{ki}\}-x_{ki}}{max_k\{x_{ki}-min_k\{x_{ki}\}\}}\right] + \frac{1}{m+S}\left[\sum_{r=1}^{S}\frac{y_{kr}-min\{y_{ki}\}}{max_k\{y_{kr}\}-min_k\{x_{kr}\}}\right]
\end{aligned}
\tag{27}
$$

For now, exact preference and equal weights do not provide insights and robust results. Due to the standard weights associated with each dimension, it is considerably hard to achieve a consensus. Therefore, this study provides an insight into the virtue of the nonparametric frontier method through an apprehensive environmental perspective. In contrast, commonly used inputs, such as capital and labor, may not be incorporated to construct the development of EVIs.

Generally, [49] excluded inputs to develop an environmental efficiency index since it generates per unit of lousy output. In this study, the wrong outputs are considered inputs as they both indicate the cost type, which implies that they follow the properties of "smaller the best." Considering the view, this study treats energy consumption as input and assumes that it would also follow the "smaller the best," considering the sustainable environment. Based on diversification indices that measure risk-free energy supplies by assuming that riskier energy supplies pose a more significant threat to energy security while at the same time reducing the energy security impact on energy efficiency and energy intensity. Therefore, risk-free energy supplies need to assess.

$$
RIES - CR_i = HHI - CR_i \times DEP_i = D_i\sum_{i=1}^{N} W_{ij}{}^2 \times CR_j
\tag{28}
$$

$$
RIES - CR_i = HHI - PE \times DEP_i = DEP_i\sum_{i=1}^{N} W_{ij}{}^2 \times \frac{1}{PE}
\tag{29}
$$

$$
RIES - PE = HHI - PE \times DEP_i = DEP_i\sum_{i=1}^{N} W_{ij}{}^2 \times \frac{1}{PE} \times CR_i
\tag{30}
$$

The RIES represents the risk in energy supplies, *CR* is the country risk, *HHI* is the Herfindahl Hirschman Index, *DEP* is the energy dependency on energy suppliers, *PE* shows potential exports, $W_{IJ} = \frac{X_{ij}}{\sum X_{ij}} X_{ij}$ represents the contribution of energy suppliers in over-all energy imports of the economy. The fourth variable is a financial indicator, i.e., gross domestic product (GDP), which shows each country's capability to produce revenue against a specific amount of GHGs emissions. Out of these

variables, GHG emissions and energy consumption are inputs, while total energy supplies and GDP are outputs. It is necessary to explain that Equations (6), (7), and (10) measure the energy efficiency, environmental index, and energy intensity, respectively, to analyze other countries or regions.

## 3. Results

This study aims to assess the existence of EKC theory in fifteen developing economies of Asia and attempts to probe the energy and environmental nexuses to view the trajectory of KEC.

### 3.1. Results of Panel ARDL

Table 2 represents the statistical summary of the dependent and independent variables of Equation (2). It shows that the values of minimum, maximum, kurtosis, standard deviation, average, and skewness of the indicator, as mentioned earlier, have reflected an improved understanding of the data and their distribution within the structure—the outcomes of the correlation matrix present in Table 3.

**Table 2.** Statistical summary.

|  | END | ENC | GW | $GW^2$ | FCF | PG |
|---|---|---|---|---|---|---|
| Mean | 2.47 | 923.66 | 3.66 | 27.86 | 12.48 | 1.62 |
| Median | 1.59 | 739.06 | 3.71 | 17.53 | 10.89 | 1.50 |
| Maximum | 14.54 | 3019.81 | 15.31 | 234.62 | 58.15 | 5.63 |
| Minimum | 0.03 | 115.70 | −14.35 | 0.00 | −48.21 | 0.14 |
| Std. Dev. | 2.30 | 665.90 | 3.80 | 33.26 | 15.95 | 0.79 |
| Jarque-Bera | 2.05 | 3.82 | 1.94 | 3.11 | 2.57 | 4.66 |
| *p*-value | 0.36 | 0.21 | 0.52 | 0.22 | 0.37 | 0.13 |

Source: authors own calculations.

**Table 3.** Correlation matrix.

|  | END | ENC | GW | $GW^2$ | FCF | PG |
|---|---|---|---|---|---|---|
| END | 1.00 |  |  |  |  |  |
| ENC | 0.63 | 1.00 |  |  |  |  |
| GW | 0.06 | −0.01 | 1.00 |  |  |  |
| $GW^2$ | −0.28 | 0.16 | 0.46 | 1.00 |  |  |
| FCF | 0.00 | −0.13 | 0.54 | 0.22 | 1.00 |  |
| PG | 0.00 | −0.06 | 0.35 | 0.36 | 0.14 | 1.00 |

Source: authors own calculations.

The correlation matrix results find that renewable energy usage (ENC) correlates with the ecological disorder (END). At the same time, all other concerned variables have a weak correlation with the ecological disorder. GDP-square variable ($GW^2$) shows the desired negative association with the ecological disorder. As the application of Panel ARDL apple concerning panel unit root, Table 4 represents the panel unit root results. According to the output, variables are stationary at different levels.

**Table 4.** Panel unit root.

| Variables | | Level | | 1st difference | | Decision |
|---|---|---|---|---|---|---|
| | | I | I & T | I | I&T | |
| END | LL & C | 0.86 | 1.68 | −1.67 | −0.23 | I(1) |
| | | −0.8 | −0.95 | -0.04 | −0.4 | |
| | | 4.1 | 2.1 | −5.73 | −4.24 | |
| | IPS | −1 | −0.98 | 0 | 0 | |
| ENC | LL & C | 2.5 | 0.03 | −3.19 | −2.50 | I(1) |
| | | −0.99 | −0.51 | 0 | 0 | |
| | | 4.75 | 1.48 | −5.22 | −3.85 | |
| | IPS | −1 | −0.93 | 0 | 0 | |
| GW | LL & C | −7.08 | −6.58 | – | – | I(0) |
| | | 0 | 0 | | | |
| | | −6.92 | −5.77 | | | |
| | IPS | 0 | 0 | | | |
| $GW^2$ | LL & C | −6.06 | −5.58 | – | – | I(0) |
| | | 0 | 0 | | | |
| | | −6.86 | −6.77 | | | |
| | IPS | 0 | 0 | | | |
| FCF | LL & C | −4.21 | −4.64 | – | – | I(0) |
| | | 0 | 0 | | | |
| | | −6.40 | −4.52 | | | |
| | IPS | 0 | 0 | | | |
| PG | LL & C | 0.34 | −10.99 | – | – | I(0) |
| | | −0.63 | 0 | | | |
| | | 10.75 | −7.78 | | | |
| | IPS | −1 | 0 | | | |

Source: authors own calculations. Note: Parentheses have Probability values.

To view the co-integration association between dependent and a set of independent variables of Equation (2), the study performs the bound test, depicted in Table 5, according to the bounds test results. F-statistics (estimated) is higher than the upper and lower critical value bunds. Thus, bounds test results accepted the co-integration of energy usage and environmental depletion alongside other demographic and economic indicators.

**Table 5.** Results of the bound test.

| Equation (1) | Bound Test Value | Df | Conclusion |
|---|---|---|---|
| END/ENC, GW, $GW^2$, FCF, PG | F-statistics = 12.96 > 3.61 Probability = (0.00) | (6, 334) 6 | Co-integration exists |

Source: authors own calculations. Note: table CI cited the unrestricted intercept, and no critical trend values of Lower bound at 5% = 2.45 and unrestricted intercept and no trend critical values of Upper bound at 5% = 3.61.

Then, Panel ARDL applied to test the long-run influence of economic upswing on environmental depletion. The Panel ARDL projection for the long run prearranges in Table 6.

**Table 6.** Panel autoregressive-distributed lag (ARDL) (Long Run).

| No. of Panels = 15 Dependent Variable = END | | | | |
| --- | --- | --- | --- | --- |
| **Regressor** | **Coefficients** | **Standard. Error** | **t-Statistics** | **_p_-Value** |
| ENC | −0.26 *** | 0.08 | 3.25 | 0.0012 |
| GW | 0.78 *** | 0.33 | 2.36 | 0.0183 |
| $GW^2$ | −0.31 ** | 0.17 | −1.82 | 0.0688 |
| FCF | 0.11 ** | 0.05 | 2.21 | 0.0271 |
| PG | 0.61 | 0.41 | 1.487 | 0.1370 |

Source: authors own calculations, Note: ** 5% and *** 1% show the statistical significance level.

The Panel ARDL findings confirmed EKC (inverted U-shape curve) for these selected developing Asian economies. Economic upswing (GW) also has a positive association in terms of ecological disorder and found that 0.78 units of $CO_2$ emissions are being generated by the economic upswing to pollute the environment. The results depict the positive influence of GDP growth on carbon dioxide emission in developing economies, evidenced by past research [50]. However, the confirmation of EKC proved by the coefficient value −0.31 of GDP-square ($GW^2$). The negative coefficient value of $GW^2$ represents the negative bond between GDP-square and carbon emission, which confirmed the reduction of carbon dioxide emission in the developing Asian economies. Thus, reducing carbon dioxide emissions due to improved economic upswing has shown the presence of inverted U-shaped EKC in developing economies and confirms the findings of [51,52].

It further finds that renewable energy usage participates negatively in carbon dioxide emissions in these developing economies. The renewable energy usage (ENC) coefficient is −0.26, which indicates that one percent of energy usage is a source of 0.26 carbon emission reduction emission. The coefficient of renewable energy usage is also significant at one percent. Hence, it proved that renewable energy usage growth helps reduce the pollution of these developing economies. The previous studies also establish the same affirmative influence of renewable energy usage on developing economies' carbon dioxide emissions [53,54].

Capital formation (FCF) has shown a role in terms of increasing $CO_2$ emissions. The results show that enhancement in capital formation has increased the environmental depletion in the developing economies. Results suggest that with ceteris paribus, a one percent increase in capital formation is a source of 0.11 percent of carbon dioxide emission, and [51,55] have shown the same evidence in their past study in which the improvement in capital formation was promoting the ecological disorder. The long-run panel ARDL does not significantly affect population growth (PG) on the ecological disorder. According to past research, population growth can be an essential indicator of any country's economy. However, it has no substantial evidence to affect the environmental conditions in the ecological disorder.

According to Table 7, the short-run results are insignificance for the economic upswing and environmental depletion in these selected developing economies. The short-run regressor consists of lag terms of the previous year. Almost all variables were found to be insignificant concerning the depletion of the environment in the preceding year. Thus, the short-run results confirmed that all economic (economic upswing, GDP-square, energy usage, fixed capital formation) and demographic indicators do not affect the carbon dioxide emission of these developing economies of Asia. This short-run analysis estimates through ECM. The coefficient value of ECMit-1 is 0.26, which shows convergence toward the long run from the short run to attain equilibrium condition. The significant negative value has proven the belongings of ECM to form the steadiness by dropping error. It estimates that the unsteadiness or errors are diminished by about 26 percent each year towards the long run from the short run, which helps attain equilibrium conditions among economic upswing, fixed capital formation, energy usage, and environmental depletion in these selected developing Asian economies. Regarding our other socio-economic variables' unemployment rate, we found it positive (as expected), but not statistically significant. Our findings are consistent with the results drawn by [51,56].

**Table 7.** Short-run panel ARDL with Error Correction Model.

| | No. of Panels = 15 Dependent Variable = END | | | |
|---|---|---|---|---|
| **Variables** | **Coefficients** | **Standard. Error** | **t-Statistics** | ***p*-Value** |
| ENC | −0.26 *** | 0.08 | 3.25 | 0.0012 |
| GW | 0.78 *** | 0.33 | 2.36 | 0.0183 |
| GW$^2$ | −0.31 ** | 0.17 | −1.82 | 0.0688 |
| FCF | 0.11 ** | 0.05 | 2.21 | 0.0271 |
| PG | 0.61 | 0.41 | 1.487 | 0.1370 |
| dENC | −0.16 * | 0.09 | 1.78 | 0.0751 |
| dGW | 0.37 * | 0.21 | 1.75 | 0.0801 |
| dGW$^2$ | −0.14 | 0.11 | 1.27 | 0.2041 |
| dFCF | 0.08 | 0.05 | 1.60 | 0.1096 |
| dPG | 0.43 | 0.29 | 1.48 | 0.1389 |
| ECM$_{it-1}$ | −0.26 ** | 0.12 | −2.17 | 0.0300 |

Source: authors own calculations, Note: * 10%, ** 5% and *** 1% show the statistical significance level.

Table 8 demonstrates the results of the augmented Dickey-Fuller (ADF) test. Generally, the average time level has no stationary series, even though all series are stationary having the first difference. If the ADF test did by using the first difference, generally, the insignificant supposition is rejected at the significance level of 1% or 5%. Consequently, this stated that the data are converted into the shape of stationary with the first difference.

**Table 8.** Unit root results of the augmented Dickey-Fuller test (ADF).

| | **With Intercept** | | | **With Trends and Intercept** | | |
|---|---|---|---|---|---|---|
| **Factor** | **k Level** | **k 1st Difference** | **k Results** | **Factor** | **k 1st Difference** | **Result** |
| END | 1.21 | 2.83 | I(1) | 1.53 | 2.32 | I(1) |
| ENC | 1.61 | 2.51 | I(1) | 2.53 | 3.52 | I(1) |
| EGW | 2.41 | 5.31 | I(1) | 2.39 | 5.43 | I(1) |
| EGW$^2$ | 2.51 | 5.74 | I(1) | 1.63 | 6.32 | I(1) |
| FCF | 1.89 | 6.00 | I(1) | 1.39 | 5.97 | I(1) |
| PG | 1.19 | 4.92 | I(1) | 2.89 | 4.78 | I(1) |

Source: Author's own calculation by using E-Views 5. ENG stands for the ecological disorder, ENC shows the energy consumption, EGW is economic growth, EGW$^2$ is the square of economic growth, FCF is capital formation, PG shows population growth.

The results of the error correction model presented in Table 9.

**Table 9.** Error correction model results.

| **Variable** | **Ln(GDP)** | **St. Error** | **t-Statistics** |
|---|---|---|---|
| END(−1) | k−0.89 | (0.32) | [−2.45] |
| ENG(−1) | k−4.21 | (1.96) | [−4.19] |
| EGW(−1) | k−42.51 | (4.32) | [−5.36] |
| EGW$^2$(−1) | k−4.56 | (1.54) | [−2.39] |
| FCF(−1) | k−5.39 | (2.34) | [−1.42] |
| PG(−1) | −3.20 | (2.730) | [−1.21] |
| C | 327.16 | – | k– |
| EC$_{t-1}$ | −0.12 *** | (0.05) | [−2.69] |

Source: Author's own calculation by using E-Views 5. Note: *** denotes 1% significance level. ENG stands for the ecological disorder, ENC shows the energy consumption, EGW is economic growth, EGW2 is the square of economic growth, FCF is capital formation, PG shows population growth.

The EC$_{t-1}$ coefficient demonstrates the short-run adjustment rate to the long-run rate, whereas the adjustment rate was noted as 12%; this implies that the 12% imbalance is corrected per year. The numerical coefficient of significance ensures the long-run causality among independent variables and dependent variables.

This study develops three indexes (environmental index (EVI), energy efficiency index (EE), and energy intensity index (EIN). The EVI index develops by using all indicators of Table 1. In contrast, the EE index utilizes energy efficiency and energy consumption as input indicators, and the EIN index has been developed by dividing energy consumption by GDP.

*3.2. Results of DEA*

Results of the DEA-based environmental index, energy index, energy intensity, and aggregated index are depicted as follows:

According to Figure 1, underline developing economies are passing through different environmental development phases compared to each other. Currently, a mixed condition is observed in this region in terms of environmental performance. Jorden has better conditions in this lineup, followed by Sri Lanka and Malaysia, while Philippine and Bangladesh are poor performers. The comprehensive set of indicators' choices are similar to the work done by [57–60].

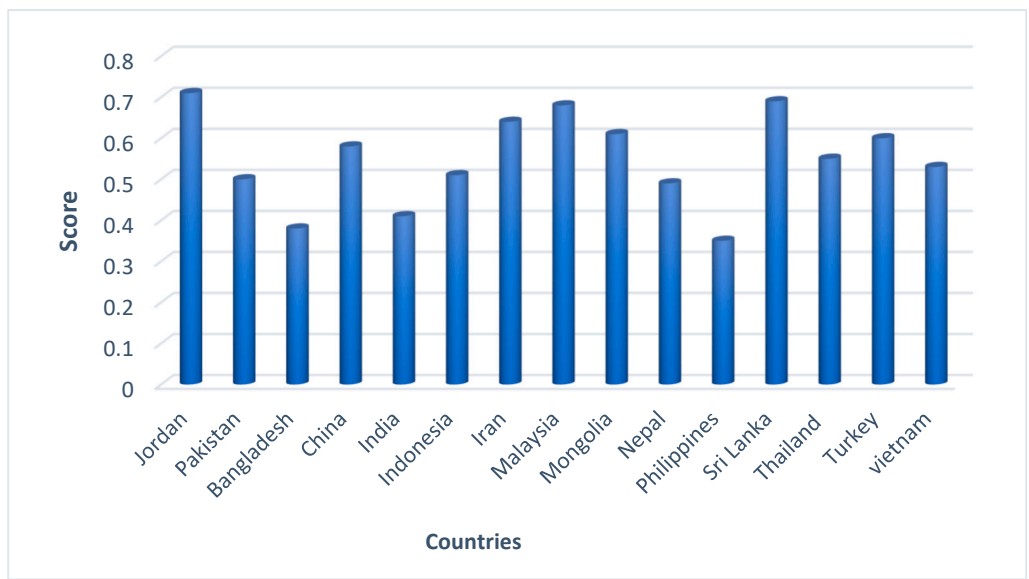

**Figure 1.** Environmental index. Source: authors' calculations.

Figure 2 shows the energy efficiency trend in these countries. China, India, and Turkey are energy efficient among these countries, while Nepal and Pakistan are the least energy-efficient countries. Thus, it may have happened due to more expenditure on China and India's renewable energy sources to meet their growing power demand for economic growth.

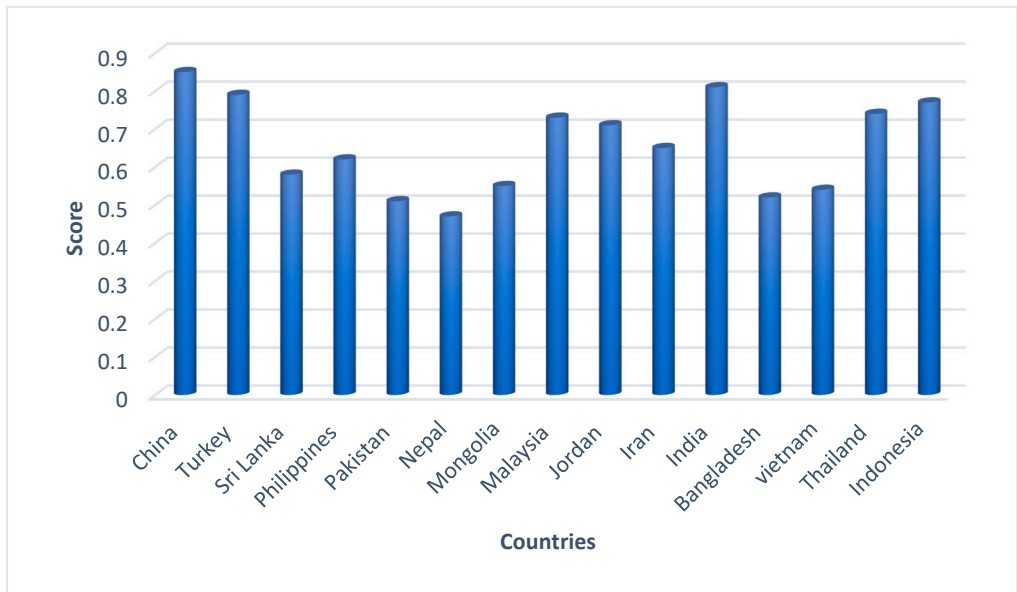

**Figure 2.** Energy efficiency index. Source: authors' calculations.

The greater energy intensity represents a higher price or cost transformed into real GDP. The level of energy intensity indicates that there is a decoupling of energy consumption and economic development. According to Figure 3, Iran is the most energy-intense economy among this dataset, followed by Nepal and Mongolia. The case of Iran's energy intensity can be valid as most energy-exporting countries find themselves in such conditions. In our results, Sri Lanka and Turkey are the least energy-intense economies. The same results are generated by [29] for the BRICS region.

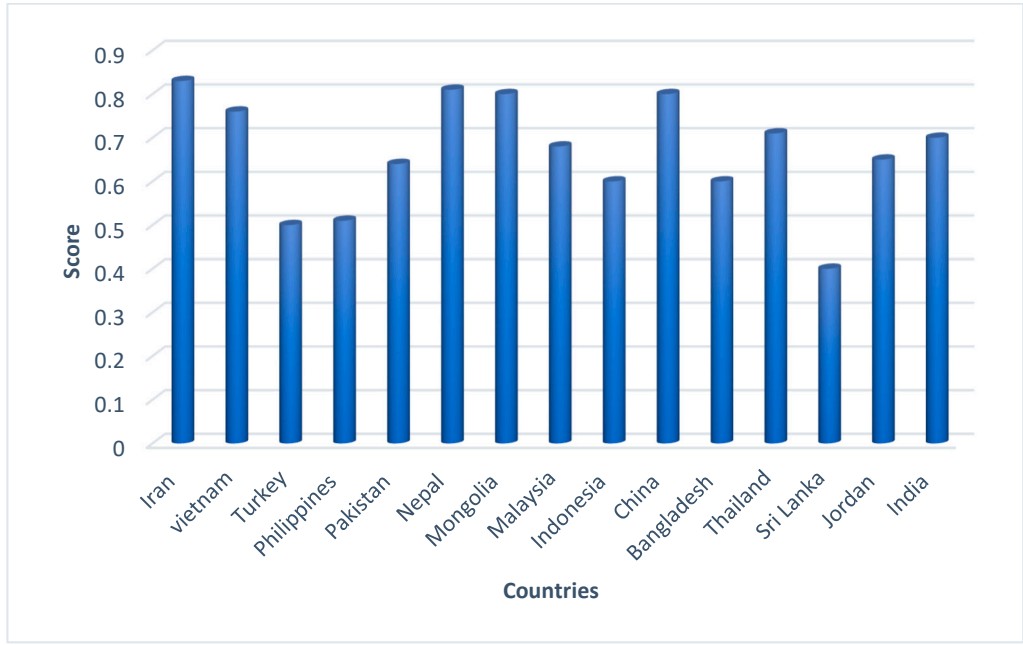

**Figure 3.** Energy intensity index. Source: authors' calculations.

Thus, all three indexes' distributions and frequencies confirm that these countries have no horizontal pattern regarding energy and environment, and there is an inconsistency between one country and the other. It may also confirm that individual initiatives matter for collective results.

Usually, decoupling expects to decrease environmental pressure from fossil-based energy production and consumption. The relationship between energy efficiency and economic factors show that energy efficiency improvements concentrate on decreasing fuel costs. However, its environmental effect relies on the nature of energy. Figure 4 shows a clearer picture of the aggregate performance of these countries.

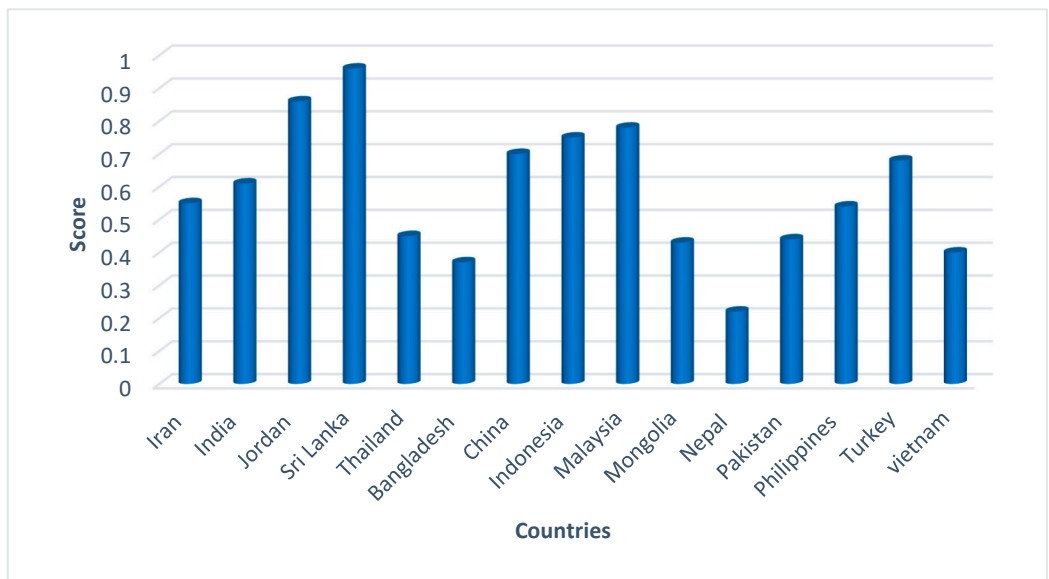

**Figure 4.** An aggregate score of all three indexes. Source: authors' calculation.

According to Figure 4., countries such as Nepal, Bangladesh, Vietnam, Pakistan, Magnolia, and Thailand are suffering to attain the best combination set of energy efficiency, energy intensity, and environmental protection. It shows that severe environmental issues (such as global warming) are linked with higher energy consumption due to rapid industrialization and urbanization [61]. Thus, renewable energy transformation can be the primary solution to enhance the energy efficiency, reducing energy intensity, and maintain sustainable environmental conditions, as also suggested by [30,62].

## 4. Discussion

This study aims to verify the existence of EKC and its trajectory with a fresh dataset of the fifteen developing economies of Asia. According to the results of panel ARDL, the theory of EKC exists in the underline developing economies. Here, the indicator of economic growth and its square term shows the positive and negative signs, respectively, EKC's confirmation statement. According to panel ARDL, the probability of EKC in these emerging economies exists at a 0.26 convergence rate in the long run. These results are in line with [27]. However, economic growth may not allow having happened as it requires energy sources, which are mostly carbon-based. Thus, the speed and trajectory of EKC heavily depend on energy sources. Therefore, energy may treat as an essential factor for EKC. The more the energy sources will be renewable and carbon-free, the more chance will be a smooth transaction towards EKC.

Most of the previous results confirm this notion for many developing economies, but these studies usually completed analysis. This study applied DEA to analyze the current condition of underline developing economies regarding energy efficiency, energy intensity, and environmental sustainability. Our DEA-based energy and environmental index results show that most of these developing economies are suffering to reduce energy intensity, increase energy efficiency, and maintain sustainable environmental conditions. The same results have been depicted by [31,32]. These results support the notion of EKC trajectory, which depended on the economic growth process and other favorable indicators, such as renewable energy usage and improvement of energy intensity with technological innovations.

The results of this study confirm that renewable energy consumption has a negative relation to carbon emission. Other than the square term of economic growth, some factors can help EKC's occurrence in these developing economies. However, these countries' energy efficiency is abysmal, which indicates that these economies depended on inefficient energy sources such as fossil fuel. Renewable energy sources could not only help in the occurrence of EKC, but they have the potential to enhance the trajectory rate of EKC as well. These findings are in line with previous studies [28,29,63].

The fixed capital formation results confirm that it also has significant features for carbon emission in these economies. It is due to the low usage of innovative technologies in these underline countries. Therefore, The EKC model shows that practical, efficient energy policies can reduce energy-based carbon dioxide emissions without damaging economic progress. For sustainable economic growth and to reduce greenhouse gas emissions, adopting clean and efficient energy sources is essential. Therefore, these developing economies' governments should focus on energy efficiency for long-term sustainable economic growth with less stress on environmental conditions. The same suggestions are made by [32,59].

Future research should ensure the results to be more general and broader. In this context, further criteria shall consider the selection of varying indicators. In applying a nonparametric frontier approach to measure the environmental vulnerability index, the study does not provide a strategy to include the decision-makers' preference weights. Therefore, further evaluation, such as on rank information and decision-makers' preferred weight, could also be included in the future.

## 5. Conclusions and Policy Recommendations

This study applied panel ARDL and DEA simultaneously to assess economic growth and energy consumption in ecological disorder. Empirical results of panel ARDL confirm the inverted U-shaped EKC for these underline emerging countries as the GDP square's coefficient is significant with a negative sign. It implies that underlined countries expect to follow the EKC theory. Renewable energy also shows the negative sign for an ecological disorder, which implies that more renewable energy use can help to mitigate carbon emission. The indexes of energy efficiency, energy intensity, and environment show that underline countries suffer from environmental conditions due to high energy intensity and low energy efficiency. As economic growth demands more and more energy supply, renewable energy is the only source to meet this demand without compromising the environment. Thus, the conversion and trajectory of EKC heavily depend on energy consumption and energy sources. According to the results mentioned above, although EKC exists for underline developing economies, the trajectory of EKC has a question surrounding what and how this will happen. Therefore, it needs to be considered the other fundamentals (such as renewable or zero-carbon energy sources) to enhance the probability of EKC theory and speedup of its trajectory.

In the wake of the COVID-19 pandemic and global economic recessions, which resulted in a drastic drop in oil and other fossil fuel prices, green energy and energy efficiency projects are losing their economic feasibility. It will endanger the achievement of the Paris agreement goals on climate change and several sustainable development goals. Based on this study's results, the policy recommendations for the developing countries are to adopt new supportive policies for the development of green energy and energy efficiency projects. The emerging economies should endorse the development and

transformation of low-carbon concepts and adopt a sustainable energy system. One of the significant obstacles to developing renewable energy and energy efficiency projects is their difficulties accessing finance. These projects are considered risky projects; hence, many financiers are reluctant to finance these projects [51,52]. Therefore, in the current and post- COVID-19 era, the necessity of employing green finance tools is highlighted [63].

Other actions can help to mitigate air pollution. For example, identifying and monitoring air pollution sources from industrial energy consumption, fuel supplies of regulating petroleum, and boosting vehicle sectors to adopt green fuels can substitute petroleum products. On the other hand, the reliance on renewable energy may enhance developing economies' growth and reduce the usage of fossil fuels. Diversification of the energy basket and relying more on renewable energy resources can also enhance energy security [51,52]. Simultaneously, increasing energy efficiency will be considered a cost-effective way to decrease energy production's environmental influences. Therefore, this study suggests that sustainable renewable energy production can be one of the main factors for sustainable development goals.

**Author Contributions:** Conceptualization, done by Q.A. and J.Z.; methodology, software form by M.M.; validation and formal analysis did by F.T.-H.; investigation, resources, data curation, performed by K.J.; writing—original draft preparation is done by M.A.; writing—review and editing and policy recommendations by W.L.; visualization, and supervision by M.A. All authors contributed equally to this paper. This research paper is contributed by the authors mentioned above in the following way. All authors have read and agreed to the published version of the manuscript.

**Funding:** This research was funded by the Zhejiang province soft science project fund (2019C35019). Farhad Taghizadeh-Hesary acknowledges the financial support of the Japan Society for the Promotion of Science (JSPS) Kakenhi (2019–2020) Grant-in-Aid for Young Scientists (19K13742), and Grant-in-Aid for Excellent Young Researcher of the Ministry of Education of Japan (MEXT).

**Conflicts of Interest:** It is submitted that the manuscript mentioned above is initially written in all aspects and submitted for the possible publication in the journal *Sustainability*. This manuscript tries to fill the gap of literature for the other essential factors, mostly renewable energy, for the smooth and quick trajectory of EKC for developing economies. We declared no conflict of interest among all authors, and they unanimously agreed to possible publication in *Sustainability*.

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
