# Peer review of "Reassessing the Environmental Kuznets Curve in Relation to Energy Efficiency and Economic Growth"

_sustainability, doi:10.3390/su12208346_

Round 1

Reviewer 1 Report

This article has used panel data (1990-2013) of 15 low/medium economy countries to analyse various relationships between economic growth, energy use and environmental impact by employing ARDL model and the DEA. The researchers highlight the importance of renewable energy technologies for sustainable economic development (i.e. growth without degrading environment).

Few suggestions:

-do not provide elementary details (e.g., the definition of climate change)

-references are missing in many places

-clear justifications (research gaps) for this research work need to be explained. For example, one of the objectives was to examine the relationship between RE and CO2 emission. For me, this is very much known fact. Give some explanations so that audience can understand the purpose of this work.

-Similarly, the researchers made straightforward and obvious statements in the conclusion section. For example “ ..more use of renewable energy can help mitigate carbon emission”;  “....energy consumption is directly responsible for global warming”. Everybody knows these facts now.

It would be helpful if the researchers highlight the contribution of this research work to the literature and industry (i.e. practical implications)

Author Response

Response to Reviewer # 1 Comments

Point 1: do not provide elementary details (e.g., the definition of climate change)

Response 1:     Thank you very much for your kind suggestions. The elementary details of climate change have been removed in the revised manuscript.

Point 2: references are missing in many places

Response 2:     The manuscript has been revisited thoroughly and the missing references against fact and figures have been added.

Point 3: clear justifications (research gaps) for this research work need to be explained. For example, one of the objectives was to examine the relationship between RE and CO2 emission. For me, this is a very much known fact. Give some explanations so that audience can understand the purpose of this work.
Response 3:     A few have tried to fix the three-dimensional effect of energy on economic growth and environmental stress. Moreover, the EKC literature focused on GDP and conversion of GDP square term but not much consider the other variables like capital formation growth rate and consumption of renewable energy. Although, these two have also independent effect on economic growth and environmental condition but can be helpful for EKC trajectory and speed. Some other studies attempted to measure the efficiency of energy in the context of economic cost and environment[33] [34]. However, they did not explain EKC to view the real impact of energy efficiency on economic growth and the environment. Thus, there is a need for some comprehensive research in these areas, especially from developing nations of Asia. These nations are suffering much in terms of ecological disorder, energy usage, and sustainable economic growth. This research attempts to cover these two different concepts. Here, the combined effect of energy, economic, and environment in the underlined developing economies analysis to understand the EKC trajectory and speed. The efficiency score of the study indicates the current condition of energy efficiency, energy intensity, and environmental efficiency of the individual countries based on the last eighteen years' progress to depict the gap of EKC among underline nations. For this purpose, lower-middle-income and upper-middle-income countries have been selected to view their respective economic and environmental conditions with energy efficiency. Therefore, the current study has novelty because of its sole combination of variables, two different angels of analysis, and the selection of nations from the Asian region concerning their income levels.

Point 4: Similarly, the researchers made straightforward and obvious statements in the conclusion section. For example “ .more use of renewable energy can help mitigate carbon emission”;  “....energy consumption is directly responsible for global warming”. Everybody knows these facts now.

Response 4:     Here, the point to consider that EKC theory can exist for these developing economies of two different income classifications, but when and how this will happen has the question mark. Therefore, it needs to be considered the other elements (such as renewable or zero-carbon energy sources) which can help to increase the chance of occurrence of EKC theory and speedup of its trajectory.

Point 5: It would be helpful if the researchers highlight the contribution of this research work to the literature and industry (i.e. practical implications)

Response 5:     Thanks for the kind suggestion. The contribution of this work in the existing literature has been added in the revised manuscript.

Reviewer 2 Report

1.In the introduction, the authors discussed the novelty of this study as "sole combination of variables and selection of nations from the Asian region concerning their income levels". The authors need to specify such aspect. Besides, the authors need to clarify the significance of such study compared to other similar ones in the field.

2.In comparison to other models, what are the reasons to use the EKC theory in this study for methodology? 

3.Under statistical considerations, how is the normality test for the data in Table 2? 

4.Under statistical considerations, how significant are the variables of END, ENC, GW, GW2, FCF and PG in Table 6 and 7? Check P values.

5.Spelling needs to be checked for the caption of Figure 1.

Author Response

Response to Reviewer # 2 Comments

Point 1: In the introduction, the authors discussed the novelty of this study as the "sole combination of variables and selection of nations from the Asian region concerning their income levels". The authors need to specify such an aspect. Besides, the authors need to clarify the significance of such a study compared to other similar ones in the field.

Response 2: Thank you very much for your kind suggestions. The necessary explanation regarding the novelty of the study has been added.

This study attempts to fill the literature gap regarding energy efficiency as the source of the EKC trajectory. Here, the combined effect of energy, economic, and environment underlined developing economies' analysis to understand the EKC trajectory and speed. The study's efficiency score indicates the current condition of energy efficiency, energy intensity, and environmental efficiency of the individual country based on the last twenty-three years progress to depict the gap of EKC among underline nations. Thus, lower-middle-income and upper-middle-income countries have been selected to view their respective economic and environmental conditions with energy efficiency as per world bank classification. Therefore, the current study has novelty because of its sole combination of variables, two different angels of analysis, and the selection of nations from the Asian region concerning their income levels. This study can help policymakers, and business individuals decide the course of EKC occurrence and its trajectory for preferring the supportive sources of renewable energy with high efficiency and low intensity in their respective countries.

Response 1: Most of the studies regarding EKC attempted to measure GDP and GDP2

Point 2: In comparison to other models, what are the reasons to use the EKC theory in this study for methodology? 

Response 2: Different studies used an ecological framework (IPAT and STIRPAT) to examine the impact of economic activities on environmental pollution. However, the present study employed EKC theory to test the nonlinearity between economic activities and carbon emissions relationships. Here, our dependent variable is carbon emissions, so the EKC theory is best performed to examine the relationship between air pollution and economic activities and to test the nonlinearity (Wagner, 2007).

  • Wagner M 2007 The carbon Kuznets curve: a cloudy picture emitted by bad econometrics? Resour. Energy Econ. 30 388–408

Point 3: Under statistical considerations, how is the normality test for the data in Table 2?

Response 3: To test the data series are from a normal distribution, we applied the Jarque-Bera test. If the p-value < 5%, it shows the rejection of the null hypothesis of normal distribution. Simultaneously, the p-value of the Jarque-Bera test for all the variables is more significant than 5%, which shows the acceptance of the null hypothesis of normal distribution. Thus, it can be concluded that the dependent and independent variables in the proposed model are following the standard distributions.      

Point 4: Under statistical considerations, how significant are the variables of END, ENC, GW, GW2, FCF, and PG in Tables 6 and 7? Check p-values.

Response 4: Following this suggestion, we have assigned a significance level by using p-value, and in Tables 6 and 7, we have generated new columns to show the p-values.

Point 5: Spelling needs to be checked for the caption of Figure 1.  
Response 5:    The spelling of the figure1 caption has been checked and rectified

Reviewer 3 Report

The manuscript is interesting and has practical value. Should be a good point to conduct similar researches on the other countries or groups of countries.

Comments and remarks:

Revision of English and the use of words are needed.

Introduction:

  • line 40 ”catastrophic tragedy” - ”catastrophe” can be defined as ”the final event of the dramatic action especially of a tragedy” - therefore, the use of these two words together it should be avoided
  • rephrase for some phrases such as line 63 ” industrialization degrades the environment” (a rephrase could be ”industrialization had led to environmental degradation”)
  • line 76 - ”Magnolia” - maybe ”Mongolia” 
  • all acronyms should be explained before being used, as the article might be read by some people who are not very familiar with the topic and the related terms (such as EKC - line 29 or GHG - line 41)
  • the correct term is CO2 emissions (not emanations....as it is used extensively in the paper)
  • Lines 94-100 should use the same tense not switch from Present simple to Present continuous and even infinitive form
  • Lines 101-103 should be rephrased.

Review of literature:

  • Lines 106-107 "Sufficient research conduct to measure the relationship between environmental problems and macroeconomic variables during the last few decades." - should be revised as it does not draw a conclusion

Materials and Methods & Results

  • "Here" should not be used as this is an academic paper
  • All Tables and Figures should have Source

Conclusions:

  • Lines 543-545 "Thus, there is a need investment to highlight the dangers of reliance on non-renewable energy resources and to convince the entrepreneurs and industrialists to use renewable energy sources to manufacture goods at a domestic level." - should be revised as it does not draw a conclusion
  • More connection between Conclusions and the conducted experiment 

Overall:

  • Page 24 - should be deleted
  • The chapters and subchapters should be renumbered 
  • The entire paper format should be changed as to have tabs or spaces to see the different paragraphs

Author Response

Response to Reviewer # 3 Comments

Point 1: Revision of English and the use of words are needed.

Response 1:     Manuscript thoroughly revisited, and issues of language have been rectified

Point 2: line 40” catastrophic tragedy” -” catastrophe” can be defined as” the final event of the dramatic action especially of a tragedy” - therefore, the use of these two words together it should be avoided.

Response 2:     The sentence has been rephrasing in the revised version as “Climate change is a devastating phenomenon that people have experienced for the last few decades”.

Point 3: rephrase for some phrases such as line 63” industrialization degrades the environment” (a rephrase could be” industrialization had led to environmental degradation”)

Response 3:     Line 63 has been rephrased as “The environment is being affected by CO2 emissions due to energy process, as industrialization had led to environmental degradation” and the issues like that have been revisited in the revised version.

Point 4: line 76 -” Magnolia” - maybe” Mongolia”

Response 4:     Thank you very much for indicating the typo mistake. Such kind of error and omissions has been rectified in the revised version

Point  5: all acronyms should be explained before being used, as the article might be read by some people who are not very familiar with the topic and the related terms (such as EKC - line 29 or GHG - line 41).

Response 5:     Thanks for the kind suggestion. The manuscript reviewed thoroughly, and all acronyms have been defined and explain adequately before their use.

Point 6: the correct term is CO2 emissions (not emanations....as it is used extensively in the paper).

Response 6:     The word “CO2 emanations” has been replaced with CO2 emission in the whole manuscript.

Point 7: Lines 94-100 should use the same tense does not switch from Present simple to Present continuous and even infinitive form.

Response 7:     Thanks for the suggestion regarding correction in the sentence. The necessary grammatical correction has been made in the revised version as per kind directions.

Point 8: Lines 101-103 should be rephrased.

Response 8:     Thanks for your suggestions. Above mentioned lines have been replaced with suitable language structure.`

Point 9: Lines 106-107 "Sufficient research conduct to measure the relationship between environmental problems and macroeconomic variables during the last few decades." - should be revised as it does not draw a conclusion.

Response 9: The sentence mentioned above has been revised in the new manuscript to enhance its meanings

Point 10: "Here" should not be used as this is an academic paper.

Response 10:   It has been replaced with appropriate word

Point 11: All Tables and Figures should have a Source.

Response 11:   All the tables and figures have been tagged with proper sources

Point 12: Lines 543-545 "Thus, there is a need investment to highlight the dangers of reliance on non-renewable energy resources and to convince the entrepreneurs and industrialists to use renewable energy sources to manufacture goods at a domestic level." - should be revised as it does not draw a conclusion.

Response 12:   This sentence has been revised and replaced with the appropriate words and language to enhance the meaning of it. The purpose of this sentence is also to draw the attention of officials and policymakers that EKC can be true to some extent but due to low transformation from short-run to long-run equilibrium and the usage of fossil fuel energy may be the hurdles which emerging economies can face. So, it is suggested to spent money and devote time to encourage entrepreneurs and businessmen to get interested in renewable energy innovation which can help a lot to the occurrence of EKC theory.

Point 13: More connection between Conclusions and the conducted experiment 

Response 13:   The conclusion has been revised according to the conducted experiment as per kind suggestions

Point 14: Page 24 - should be deleted

Response 14:   page 24(last page of MDPI format example) has been removed as per the direction

Point 15: The chapters and subchapters should be renumbered.

Response 15:   The chapters and subchapters has been revisited after the revision of the manuscript 

Point 16: The entire paper format should be changed as to have tabs or spaces to see the different paragraphs.

Response 16:   The paper format has been rechecked and rectified as per journal as well as kind directions.